# Research on Anisotropic Viscoelastic Constitutive Model of Compression Molding for CFRP

**DOI:** 10.3390/ma13102277

**Published:** 2020-05-15

**Authors:** Jiuming Xie, Shiyu Wang, Zhongbao Cui, Jin Wu, Xuejun Zhou

**Affiliations:** 1School of Mechanical Engineering, Tianjin University, Tianjin 300350, China; wangshiyu@tju.edu.cn; 2School of Mechanical Engineering, Tianjin Sino-Germen University of Applied Sciences, Tianjin 300350, China; zhouxuejun@tsguas.edu.cn; 3Tianjin Sinotech Industry Co., Ltd., Tianjin 300350, China; cuizhongbao@126.com

**Keywords:** CFRP, compression molding, constitutive model, TSMs, temperature stiffness coefficient, TCMs

## Abstract

The carbon-fiber-reinforced polymer (CFRP) is a mainstream material for lightweight products from the end of the 20th century to the present day. Its compression molding process has obvious advantages in mass production. This paper attempts to establish the constitutive models of compression molding of the CFRP materials and study their mechanism. Based on anisotropic linear elastic mechanics, viscoelastic mechanics, and thermodynamics, as well as the Maxwell viscoelastic constitutive model, we first establish the constitutive model of thermorheologically simple CFRP materials (TSMs). Then, considering the influence of temperature on the initial stiffness and equilibrium stiffness, the concept of temperature stiffness coefficient is introduced, and the Cartier coordinate system is converted into a cylindrical coordinate system, thereby establishing the constitutive model of thermorheologically complex materials (TCMs) using the tensor form. Finally, by comparing to the structure of the Zocher model, the two constitutive models established in this study are verified. The research findings have important theoretical research significance for studying the compression molding mechanism of carbon fiber and further improving the quality of product molding.

## 1. Introduction

Carbon fiber is one of the most relevant materials from the end of the 20th century to the present day. Due to its characteristics such as high specific strength, high specific modulus, fatigue resistance, good molding process, good breakage safety, and strong performance designability, the carbon-fiber-reinforced polymer (CFRP) has been increasingly applied in aerospace, wind turbine blades, sports equipment, and automotive parts, etc. It is the mainstream material for lightweight products in the 21st century [1,2,3], gradually transiting from secondary load-bearing components to primary load-bearing components.

Commonly used molding methods of the CFRP materials include compression molding, autoclave, winding, and the squeezing method, etc. Among them, the compression molding method has the advantages of low cost, high efficiency, low internal stress, small warpage, good mechanical stability, and high repeatability, and it has a strong competitive advantage in the batch production of product parts, especially in large batch production [4,5]. However, the compression molding process is disturbed by multiple physical fields, multiple process parameters, the characteristics of the material itself, the thermochemical reaction of the substrate, etc., resulting in residual stress during the molding process of the product, which directly affects the curing deformation of the material [6,7]. Therefore, the constitutive model of the CFRP molding process should be established. It is of great significance for reasonably predicting the residual stress development of the product molding process, optimizing the molding process, and improving product quality.

Scholars worldwide have conducted a lot of research based on the constitutive model of the CFRP materials and achieved rich results. The linear elastic constitutive equation combined with laminate theory is a commonly used method to predict the residual stress of composite materials. Stango et al. [8] assumed that the material properties during the curing process were constants and adopted the laminate theory to study the temperature-induced composite residual stress. Bogetti et al. [9] believed that the performance of the composite material was related to the curing degree and proposed an improved linear elastic constitutive model, namely the CHILE (α) model. Johnston [10] stated that the performance of the composite material during curing is related to the glass transition temperature Tg of the matrix, and they established the CHILE ($t_{g}$) model. In essence, both the CHILE (α) and CHILE ($t_{g}$) models are linear elastic constitutive models, which cannot reflect the characteristics of material stress relaxation and creep; the CFRP resin matrix exhibits obvious viscoelastic properties at high temperatures. Thus, the viscoelastic constitutive model can truly describe the mechanical properties of composite materials during curing. Zocher et al. [11] used the generalized Maxwell model and Prony technology to express the relaxation stiffness of composite materials, and they derived the anisotropic constitutive equation and incremental equation. Kim et al. [12,13] assumed that Poisson’s ratio of the resin during the curing process was unchanged, and combined the time–temperature-curing degree equivalence principle to obtain the viscoelastic properties of related resins, which provided important references for understanding the viscoelastic behavior of the materials during curing. Zobeiry et al. [14] used a differential type to represent the viscoelastic constitutive equation and its increment. Abouhamzeh et al. [15] combined the Laplace transform and inverse transform and proposed a solution to predicting the viscoelastic behavior of anisotropic composite materials. Svanberg et al. [16] proposed a path-dependent constitutive model and studied the curing and post-curing deformation mechanism of L-shaped parts.

Based on the previous research, this paper establishes the CFRP anisotropic viscoelastic constitutive models of thermorheologically simple CFRP materials (TSMs) and thermorheologically complex materials (TCMs) by combining with anisotropic linear elastic mechanics, viscoelastic theory, and thermodynamics, and the Maxwell viscoelastic model. Considering the effect of material temperature on the stiffness and equilibrium stiffness of the carbon fiber molding process, the temperature-dependent thermoelastic stiffness coefficient was introduced, and the Cartesian coordinate system was converted into a cylindrical coordinate system, thereby establishing the CFRP anisotropic elastic constitutive equation and increment equations of the TCM materials. The establishment of the CFRP anisotropic viscoelastic constitutive models, especially for TCM materials, will lay a theoretical foundation for revealing the mechanism of carbon-fiber-reinforced polymers [17], and then provide theoretical guidance to improve the quality of products molding, to further speed up the industrialization process of CFRP.

## 2. Theoretical Basis of CFRP Compression Molding

Generally, if the performance of each point in the object is the same, it is a homogeneous material, otherwise, a heterogeneous material. In addition, materials that show the same properties in every direction at every point in the object are called isotropic materials, while those with different properties at every point in the object are anisotropic materials. For a CFRP material, the matrix is an isotropic material, and the reinforcing fiber is a transversely isotropic body, so the CFRP material is an anisotropic material. In the molding process of CFRP materials, the matrix material has undergone a transition process from the viscous fluid state to rubber state to glass state, etc. It is necessary to combine the elastic mechanics of the composite material and the viscoelastic mechanics in the research.

### 2.1. The basis of Anisotropic Linear Elastic Mechanics

To study the mechanical properties of composite materials and solve the stress and strain of a continuous elastomer under external load, we need to balance the equations, geometric equations, constitutive equations, etc.

In the space coordinate system xyz, the stress state of any point in the elastic body under any load is represented by the normal stress components $\sigma_{x}$, $\sigma_{y}$, and $\sigma_{z}$, and the shear stress components $\tau_{xy}$, $\tau_{yz}$, and $\tau_{zx}$. According to the theorem of conjugate shearing stress, there is $\tau_{xy}=\tau_{yx}$, $\tau_{yz}=\tau_{zy}$, and $\tau_{zx}=\tau_{xz}$. $F_{x}$, $F_{y}$, and $F_{z}$ respectively represent the external force components received by the elastic body in the x, y, and z directions. Ignoring the volume force, the equilibrium relationship of any point in the elastic body along the coordinate axes x, y, and z is:(1)$$\left\{ \begin{array}{l} \frac{\partial \sigma_{x}}{\partial x}+\frac{\partial \tau_{\mathrm{yx}}}{\partial y}+\frac{\partial \tau_{\mathrm{zx}}}{\partial z}+F_{x}=0\\ \frac{\partial \tau_{\mathrm{xy}}}{\partial x}+\frac{\partial \sigma_{y}}{\partial y}+\frac{\partial \tau_{\mathrm{zy}}}{\partial z}+F_{y}=0\\ \frac{\partial \tau_{\mathrm{xz}}}{\partial x}+\frac{\partial \tau_{\mathrm{yz}}}{\partial y}+\frac{\partial \tau_{z}}{\partial z}{+F}_{z}=0 \end{array}\right.$$

Similarly, in the space coordinate system xyz, the strain state of any point on the elastic body can be represented by the normal strain components $\varepsilon_{x}$, $\varepsilon_{y}$, and $\varepsilon_{z}$ and the shear strain components $\gamma_{xy}$, $\gamma_{yz}$, and $\gamma_{zx}$ at that point. If u, v, and w represent the displacement components in the three directions of *x*, *y*, and *z*, then the geometric equation of any point in the elastic body along the x, y, and z directions is:(2)$$\left\{ \begin{array}{l} \varepsilon_{x}=\frac{\partial u}{\partial x},\gamma_{yz}=\frac{\partial \omega }{\partial y}+\frac{\partial v}{\partial z}\\ \varepsilon_{y}=\frac{\partial v}{\partial y},\gamma_{xz}=\frac{\partial u}{\partial z}+\frac{\partial \omega }{\partial x}\\ \varepsilon_{z}=\frac{\partial \omega }{\partial z},\gamma_{xy}=\frac{\partial v}{\partial x}+\frac{\partial u}{\partial y} \end{array}\right.$$

According to Equations (1) and (2), in the space coordinate system xyz, there are 15 unknown functions, namely 6 stress components, 6 strain components, and 3 displacement components. Their relationship is shown as:(3)$$\left\{\begin{array}{l} \sigma_{x}\\ \sigma_{y}\\ \sigma_{z}\\ \tau_{yz}\\ \tau_{zx}\\ \tau_{xy} \end{array}\right\}=\left\{\begin{array}{l} C_{11}\;C_{12}\;C_{13}\;C_{14}\;C_{15}\;C_{16}\\ C_{21}\;C_{22}\;C_{23}\;C_{24}\;C_{25}\;C_{26}\\ C_{31}\;C_{32}\;C_{33}\;C_{34}\;C_{35}\;C_{36}\\ C_{41}\;C_{42}\;C_{43}\;C_{44}\;C_{45}\;C_{46}\\ C_{51}\;C_{52}\;C_{53}\;C_{54}\;C_{55}\;C_{56}\\ C_{61}\;C_{62}\;C_{63}\;C_{64}\;C_{65}\;C_{66} \end{array}\right\}\left\{\begin{array}{l} \varepsilon_{x}\\ \varepsilon_{y}\\ \varepsilon_{z}\\ \gamma_{yz}\\ \gamma_{xz}\\ \gamma_{xy} \end{array}\right\}$$

Among them, $C_{mn}$(m,n = 1, 2, 3, 4, 5, 6) is the stiffness coefficient.

The matrix of the stiffness coefficient is symmetric, and only 21 stiffness coefficients are independent. Equation (3) establishes the relationship between stress and strain, which is called the generalized Hooke’s law or elastic constitutive equation.

For the matrix of a CFRP material, the number of independent stiffness coefficients is 2, and its stress–strain relationship is
(4)$$\left\{\begin{array}{l} \sigma_{x}\\ \sigma_{y}\\ \sigma_{z}\\ \tau_{yz}\\ \tau_{zx}\\ \tau_{xy} \end{array}\right\}=\left\{\begin{array}{l} C_{11}\;C_{12}\;C_{12}\;\;\;\;\;0\;\;\;\;\;\;0\;\;\;\;\;0\\ C_{12}\;C_{11}\;C_{12}\;\;\;\;\;0\;\;\;\;\;\;0\;\;\;\;\;0\\ C_{12}\;C_{12}\;C_{11}\;\;\;\;\;0\;\;\;\;\;\;0\;\;\;\;\;0\\ \;0\;\;\;\;0\;\;\;\;0\;2\left(C_{11}-C_{12}\right)\;0\;\;\;\;\;0\\ \;0\;\;\;\;0\;\;\;\;0\;\;\;\;\;0\;2\left(C_{11}-C_{12}\right)\;0\\ \;0\;\;\;\;0\;\;\;\;0\;\;\;\;\;0\;\;\;\;\;0\;2\left(C_{11}-C_{12}\right) \end{array}\right\}\left\{\begin{array}{l} \varepsilon_{x}\\ \varepsilon_{y}\\ \varepsilon_{z}\\ \gamma_{yz}\\ \gamma_{xz}\\ \gamma_{xy} \end{array}\right\}$$

The reinforced fiber has 5 independent stiffness coefficients. Assuming that the xoy coordinate plane is isotropic, its stress–strain relationship is
(5)$$\left\{\begin{array}{l} \sigma_{x}\\ \sigma_{y}\\ \sigma_{z}\\ \tau_{yz}\\ \tau_{zx}\\ \tau_{xy} \end{array}\right\}=\left\{\begin{array}{l} C_{11}\;C_{12}\;C_{13}\;0\;\;\;0\;\;\;\;0\\ \;\;\;\;\;\;C_{11}\;C_{23}\;0\;\;\;0\;\;\;\;0\\ \;\;\;\;\;\;C_{33}\;0\;\;\;0\;\;\;\;0\\ \;\;\;\;\;\;\;\;C_{44}\;\;\;0\;\;\;\;0\\ \;\;symmetric\;\;\;\;C_{44}\;0\\ \;\;\;\;\;\;\;\;\;\frac{1}{2}\left(C_{11}-C_{12}\right) \end{array}\right\}\left\{\begin{array}{l} \varepsilon_{x}\\ \varepsilon_{y}\\ \varepsilon_{z}\\ \gamma_{yz}\\ \gamma_{xz}\\ \gamma_{xy} \end{array}\right\}$$

### 2.2. The Basis of Anisotropic Viscoelastic Mechanics

The mechanical properties of viscoelastic materials such as shear modulus, loss modulus, and loss factor, etc. are usually related to ambient temperature, vibration frequency, strain amplitude, etc. Therefore, the constitutive relationship of viscoelastic materials is very complicated.

When describing the viscoelastic behavior of materials, the generalized Maxwell model is generally used, which consists of multiple Maxwell elements in parallel with a spring, as shown in Figure 1:

Each Maxwell element contains a spring with elastic modulus E and a viscous strain damper with strain viscosity η. In the generalized Maxwell model, the modulus is used to measure the performance of the spring, and the relaxation time is used to express the performance of the damper. The constitutive equation of anisotropic viscoelastic materials [18,19,20,21] is:(6)$$\sigma_{i}\left(t\right)=\int_{-\infty }^{t}C_{ij}\left(\alpha ,T,t-\tau \right)\frac{\partial \varepsilon_{j}^{\mathrm{eff}}\left(\tau \right)}{\partial \tau }d\tau \;i,j=1\sim 6$$
where $C_{ij}$ is the stiffness matrix, $\varepsilon_{j}^{eff}$ is the effective strain tensor, and $\sigma_{i}$, α, and τ are the stress tensor, curing degree, and virtual time integral variable of the tensor, respectively.

The effective strain tensor can be expressed by the total strain tensor $\varepsilon_{j}$ and the non-mechanical strain tensor $\varepsilon_{j}^{tc}$, namely:(7)$$\varepsilon_{j}^{eff}\left(\tau \right)=\varepsilon_{j}\left(\tau \right)-\varepsilon_{j}^{tc}\left(\tau \right)$$

If the material exhibits simple thermo-rheological properties when cured, the above equation can be changed to:(8)$$\sigma_{i}\left(t\right)=\int_{-\infty }^{t}C_{ij}\left(\alpha =\alpha_{0},\xi^{t}-\xi^{\tau }\right)\frac{\partial }{\partial \tau }\left[\varepsilon_{j}\left(\tau \right)-\varepsilon_{j}^{tc}\left(\tau \right)\right]d\tau \;i,j=1∼6$$
where ξ is the time and is expressed as:(9)$$\xi^{t}=\int_{0}^{t}\frac{ds}{\alpha_{T}\left(\alpha_{0},T\left(s\right)\right)}$$
(10)$$\xi^{\tau }=\int_{0}^{\tau }\frac{ds}{\alpha_{T}\left(\alpha_{0},T\left(s\right)\right)}$$

Among them, $\alpha_{T}$ is the conversion equation, related to temperature and curing degree; s is the time integral variable.

The stiffness matrix $C_{ij}$ is represented by the Prony series:(11)$$C_{ij}\left(\alpha ,\xi \right)=C_{ij}^{\infty }\left(\alpha \right)+\sum_{m=1}^{n}C_{ij}^{m}\left(\alpha \right)exp\left[-\frac{\xi \left(\alpha ,T\right)}{\tau_{m}\left(\alpha \right)}\right]\;i,j=1\sim 6$$
where $C_{ij}^{\infty }$ is the stiffness matrix after the material is completely relaxed, that is, the equilibrium stiffness; $C_{ij}^{m}$ is the stiffness matrix of each m branch in the Prony series; $\tau_{m}$ is the discrete relaxation time of the m-th branch.

### 2.3. The Basis of Thermodynamics

During the molding process of the composite material, the matrix undergoes a curing reaction to release heat, which is regarded as an internal heat source and controls the entire heat transfer process together with the applied temperature.

According to the Fourier heat conduction equation, the heat transfer process can be expressed as:(12)$$\rho C_{p}\frac{\partial T}{\partial t}=\frac{\partial }{\partial x}\left(k_{x}\frac{\partial T}{\partial x}\right)+\frac{\partial }{\partial y}\left(k_{y}\frac{\partial T}{\partial y}\right)+\frac{\partial }{\partial z}\left(k_{z}\frac{\partial T}{\partial z}\right)+Q$$
where ρ, $C_{p}$, $k_{x}$, $k_{y}$, and $k_{z}$ respectively represent the density, specific heat, and anisotropic thermal conductivity of the composite material; T is the temperature at time t; Q is the heat generated by the curing reaction of the resin, which can be calculated as
(13)$$\frac{\partial Q}{\partial t}=\rho_{m}\left(1-V_{f}\right)H_{T}\frac{da}{dt}$$

In Equation (13), $\rho_{m}$ is the resin matrix density, $V_{f}$ is the fiber volume fraction, $H_{t}$ is the total heat released by the resin during complete curing, and $\frac{d\alpha }{dt}$ is the instant curing rate of the resin. The curing degree can be expressed as
(14)$$a\left(t\right)=\int_{0}^{t}\frac{da}{dt}$$

The boundary conditions of the heat conduction model are:(15)$$K_{eff}\frac{\partial T}{\partial n}{+h}_{eff}\left(T_{s}-T_{m}\right)=0$$
where $T_{s}$ and $T_{m}$ are the surface temperature and heating temperature of the composite material, respectively; $K_{eff}$ and $h_{eff}$ are the equivalent thermal conductivity and equivalent convection heat transfer coefficients of the composite material surface, respectively.

## 3. CFRP Constitutive Model of the TSMs 

The viscoelastic constitutive modulus can reflect the creep and relaxation characteristics of the material. In this paper, the generalized Maxwell viscoelastic constitutive equation was used to reflect the viscoelastic characteristics of the simple thermal-rheological composite material.

### 3.1. The Constitutive Model Based on Viscoelastic Theory

For a single Maxwell element, the strain of the element is a combination of spring and viscous element strain, which is expressed as
(16)$$\frac{d\varepsilon }{dt}=\frac{1}{E_{0}}\frac{d\sigma }{dt}+\frac{\sigma }{\eta }$$

Using the relaxation modulus to express the relaxation characteristics of the material,
(17)$$E\left(t\right)=E_{0}{exp}^{-t/\tau }$$
where τ=η/E, i.e., the relaxation time.

For the generalized Maxwell model, the stress relaxation modulus is given as
(18)$$E\left(t\right)=\sum_{m=1}^{N+1}E_{m}{exp}^{-t/\tau_{m}}$$

$E_{m}$ and $\tau_{m}$ are the elastic modulus and relaxation time of the m-th branch, respectively.

It can be seen from Figure 1 that the N + 1-th unit has only one spring, the relaxation modulus of the viscous element is negligible, and its elastic modulus is the modulus $E^{\infty }$ after complete relaxation. The initial modulus of the generalized Maxwell model is $E^{0}$, and it can then be expressed as
(19)$$E\left(t\right)=E^{\infty }+\sum_{m=1}^{N}W_{m}\left(E^{0}-E^{\infty }\right){exp}^{-t/\tau_{m}}$$
where $W_{m}$ is the weight coefficient, which is expressed as
(20)$$W_{m}=\frac{E^{m}}{E^{0}-E^{m}}$$

The stress of the m-th branch in the generalized Maxwell model at time t is expressed as
(21)$$\frac{d\sigma_{m}^{t}}{dt}+\frac{1}{\tau_{m}^{t}}\sigma_{m}^{t}=C_{m}\frac{d\varepsilon_{eff}^{t}}{dt}$$

Its difference form is
(22)$$\frac{\sigma_{m}^{t}-\sigma_{m}^{t-\Delta t}}{\Delta t}+\frac{1}{\tau_{m}^{t}}\sigma_{m}^{t}=C_{m}\frac{\varepsilon_{eff}^{t}-\varepsilon_{eff}^{t-\Delta t}}{\Delta t}$$
where $\sigma_{m}^{t-\Delta t}$ and $\sigma_{m}^{t}$, $\varepsilon_{eff}^{t-\Delta t}$, and $\varepsilon_{eff}^{t}$ represent the stress tensor and effective strain tensor before and after the time increment Δt, respectively, $t_{m}^{t}$ and $C_{m}$, respectively represent the relaxation time and stiffness matrix of the m-th branch of the Maxwell model at time t. The effective stress tensor can be expressed as
(23)$$\varepsilon_{eff}=\varepsilon_{s}-K_{cte}\cdot \Delta t-K_{ccs}\cdot \Delta \alpha$$

Among them, $\varepsilon_{s}$ represents the total stress tensor, $K_{cte}$ and $K_{ccs}$ represent the effective thermal expansion coefficient and chemical shrinkage coefficient, respectively, Δt is the temperature increment, and Δα is the curing degree increment.
(24)$$\sigma_{m}=\frac{1}{1+\left(\Delta t/\tau_{m}^{t}\right)}\left[C_{m}\left(\varepsilon_{eff}^{t}-\varepsilon_{eff}^{t-\Delta t}\right)+\sigma_{m}^{t-\Delta t}\right]$$

Combining the stresses of all branches of the Maxwell model and the balance spring, we obtain the total stress at time t:(25)$$\sigma^{t}{=C}_{ij}\varepsilon_{eff}^{t}+\sum_{m=1}^{n}\sigma_{m}^{t}{=C}_{ij}\varepsilon_{eff}^{t}+\sum_{m=1}^{n}\frac{C_{ij}\left(\varepsilon_{eff}^{t}-\varepsilon_{eff}^{t-\Delta t}\right)+\sigma_{m}^{t-\Delta t}}{1+\left(\Delta t/\tau_{m}^{t}\right)}$$

Based on the above, the stress increment at time t can be expressed as
(26)$$\Delta \sigma^{t}=\sigma^{t}-\sigma^{t-\Delta t}{=C}_{ij}\left(\varepsilon_{eff}^{t}-\varepsilon_{eff}^{t-\Delta t}\right)+\sum_{m=1}^{n}\frac{C_{m}\left(\varepsilon_{eff}^{t}-\varepsilon_{eff}^{t-\Delta t}\right)-\left(\Delta t/\tau_{m}^{t}\right)+\sigma_{m}^{t-\Delta t}}{1+\left(\Delta t/\tau_{m}^{t}\right)}$$
where $C_{ij}$ represents the stiffness matrix of the spring, and the stiffness matrix of the m-th branch can be expressed as
(27)$$C_{ij}=\left(C_{ij}^{0}-C_{ij}^{\infty }\right)W_{m}exp\left[\frac{-\xi \left(\alpha ,T\right)}{\tau_{m}\left(\alpha \right)}\right]$$

$C_{ij}^{0}$ is the initial stiffness matrix and $C_{ij}^{\infty }$ is the stiffness matrix after complete relaxation. Generally, $C_{ij}^{\infty }=rC_{ij}^{0}$; r is a constant, and different material systems have different values of *r*.

The conversion equation can be expressed as
(28)$$log\left(\alpha_{T}\right)=\left[-\alpha_{1}exp\left(\frac{1}{\alpha -1}\right)-\alpha_{2}\right]\left(T-T_{t}\right)$$

Among them, $\alpha_{1}$ and $\alpha_{2}$ are constants, and different matrices have different values.
(29)$$\tau_{m}^{t}=\alpha_{T}\left[\alpha ,T\left(s\right)\right]\tau_{m}\left(\alpha \right)$$
(30)$$log\left(T_{m}\left(\alpha \right)\right)=log\left(\tau_{m}\left(\alpha_{t}\right)\right)\left[f\left(\alpha \right)-\left(\alpha -\alpha_{t}\right)log\left(\lambda_{m}\right)\right]$$

### 3.2. CFRP Viscoelastic Constitutive Equation and Incremental Equation of the TSMs

When the CFRP materials are TSMs, the material’s equilibrium stiffness and initial stiffness are unrelated with the degree of curing. According to Equations (11), (19), and (27), the relaxation stiffness of the CFRPs can be expressed as:(31)$$C_{ij}\left(\xi \right)=C_{ij}^{\xi }=C_{ij}^{\infty }+\left(C_{ij}^{0}-C_{ij}^{\infty }\right)\sum_{m=1}^{N}W_{m}exp\left(-\frac{\xi }{\tau_{m}}\right)$$

For anisotropic materials, the internal stress is expressed as: [13,22,23]
(32)$$\sigma_{i}^{t}=\sigma_{i}\left(t\right)=\int_{ij}^{t-t^{\prime }}C_{ij}\left(t-t^{\prime }\right)\frac{\partial \varepsilon_{j}}{\partial t}dt$$

Considering the effects of temperature T and curing degree on relaxation stiffness, the above equation can be expressed as:F
(33)$$\sigma_{i}^{t}=\int_{ij}^{t-t^{\prime }}C_{ij}\left(\xi -\xi^{\prime }\right)\frac{\partial \varepsilon_{j}}{\partial t}dt$$

Considering the stress increment $\Delta \sigma_{i}^{\Delta t}$ within the time increment Δt, there is
(34)$$\Delta \sigma_{i}=\sigma_{i}^{t}-\sigma_{i}^{t-\Delta t}$$

When the time increment is small, the curing degree is considered to be approximately constant, that is,
(35)$$\frac{\partial \varepsilon_{i}}{\partial \xi }=\frac{\Delta \varepsilon_{i}}{\Delta \xi }\;\left(\xi^{t-\Delta t}\leq \xi \leq \xi^{t}\right)$$

Therefore, the stress increment can be expressed as: [24]
(36)$$\Delta \sigma_{i}=\left(C_{ij}^{*}\right)\Delta \varepsilon_{i}^{t}-\Delta \sigma_{i}^{*}$$
(37)$$C_{ij}^{*}=C_{ij}^{\infty }+\sum_{m=1}^{N}\left(C_{ij}^{0}-C_{ij}^{\infty }\right)W_{n}\frac{\alpha_{T}\tau_{m}}{\Delta t}\left[1-exp\left(-\frac{\Delta \xi^{t}}{\tau_{m}}\right)\right]$$
(38)$$\Delta \sigma_{i}^{*}=\sum_{m=1}^{N}S_{i_{m}}^{t}\left[1-exp\left(-\frac{\Delta \xi^{t}}{\tau_{m}}\right)\right]$$
where $S_{i_{m}}$ is a historical state variable; the initial value is 0, which can be expressed as:(39)$$S_{i_{m}}^{t}=exp\left(-\frac{\Delta \xi^{t}}{\tau_{m}}\right)S_{i_{m}}^{t-\Delta t}+\left(C_{ij}^{0}-C_{ij}^{\infty }\right)W_{n}\frac{\alpha_{T}\tau_{m}\Delta \varepsilon_{j}^{t}}{\Delta t}\left[1-exp\left(-\frac{\Delta \xi^{t}}{\tau_{m}}\right)\right]$$

## 4. CFRP Constitutive Model of the TCMs

### 4.1. Thermoelastic Expressions of the TCMs

To simulate the high-temperature viscoelastic behavior of composite materials, it is generally assumed that the composite material is a simple thermal-rheological material, that is, the equilibrium stiffness and initial stiffness of the material are related to temperature and curing degree. Many experiments have shown that the initial stiffness and equilibrium stiffness are related to the material temperature [25]. In this paper, the viscoelastic materials whose equilibrium stiffness and initial stiffness change with temperature are called thermorheologically complex materials.

In the curing process of composite materials, the molding temperature and the curing degree of the matrix change with time, and the elastic modulus of the material is related to the temperature and the curing degree. On the one hand, it affects the conversion factor and relaxation time, similar to the performance of the viscous elements; on the other hand, it also affects $C_{ij}^{0}$ and $C_{ij}^{\infty }$, which is similar to the elastic properties.

When constructing the viscoelastic model of the material in this paper, the vertical movement coefficient representing the correlation between $C_{ij}^{\infty }$ and temperature was added to be the thermal viscosity stiffness coefficient. Through the experimental method and data processing method provided by Kim [12,13], the exponential function was used to express the thermoelastic stiffness coefficient of an epoxy resin base, and linearly fit it, as shown in Figure 2 below.

### 4.2. Conversion of Coordinate Systems

For a better description in the study of the CFRP mechanical properties, it is necessary to establish a local coordinate system for the geometric elements or convert the global coordinate system. In this paper, the Cartesian coordinate axis was converted into a cylindrical coordinate system for reducing the amount of calculation.

In the Cartesian coordinate system (x,y,z), the unit base vectors of the coordinate axes are $\vec{i}$, $\vec{j}$, and $\vec{k}$, respectively, and the unit vectors converted into the cylindrical coordinate system are ${\vec{e}}_{r}$, ${\vec{e}}_{\varphi }$, and ${\vec{e}}_{z}$, respectively. It is shown as
(40)$$\left\{ \begin{array}{l} \vec{i}={\vec{e}}_{r}cos\varphi -{\vec{e}}_{\varphi }sin\varphi \\ \vec{j}={\vec{e}}_{r}sin\varphi +{\vec{e}}_{\varphi }cos\varphi \\ \vec{k}={\vec{e}}_{z} \end{array}\right.$$

Then, the conversion formula in the Cartesian coordinate system and cylindrical coordinate system is:(41)$$\left\{\begin{array}{l} x_{1}\\ x_{2}\\ x_{3} \end{array}\right\}=\left[\begin{array}{l} \;\;cos\varphi \;\;sin\varphi \;0\\ -sin\varphi \;cos\varphi \;0\\ \;\;0\;\;\;\;0\;\;\;1 \end{array}\right]\left\{\begin{array}{l} x\\ y\\ z \end{array}\right\}=\beta \left\{\begin{array}{l} x\\ y\\ z \end{array}\right\}$$

The stress vector in the cylindrical coordinate system can be expressed by the stress vector $\sigma_{i}$ in the Cartesian coordinate system, namely:(42)$$\sigma_{c}=\beta \sigma_{i}\beta^{T}$$
where $\beta \beta^{T}=1$.

### 4.3. CFRP Viscoelastic Constitutive Equation and Enhancement Equation of the TCMs

As the anisotropic viscoelastic constitutive model of TCMs contains many factors, the tensor was used in this paper to express the three-dimensional viscoelastic constitutive model and incremental equations of TCMs.

For the one-dimensional Maxwell element, the stress relaxation under unit constant strain is equal to the relaxation modulus of the model. In view of the thermoelastic stiffness coefficient, the relaxation modulus of the element at time *t* is:(43)$$E\left(t\right)=\frac{E_{0}\left(T\right)}{E_{0}\left(T\left(0\right)\right)}E\left(\xi \right)=dE\left(\xi \right)$$
where d is the thermoelastic stiffness coefficient, $d=E_{0}\left(T\right)/E_{0}\left(T\left(0\right)\right)$.

$E_{0}\left(T\right)$ is the spring stiffness and $E_{0}\left(T\left(0\right)\right)$ is the spring stiffness at *t* = 0.

For the three-dimensional generalized Maxwell model, the Cartier coordinate system was converted into a cylindrical coordinate system to express the relaxation modulus. At this time, it appeared to be isotropic on the xoy plane. When calculating, the three-dimensional generalized model is equivalent to the two-dimensional generalized model, which can be compared to the relaxation modulus representation of a one-dimensional Maxwell element,
(44)$$C_{\varphi kl}^{t}=\underset{no\;sum\;on\;\varphi ,k\;and\;l}{\underset{︸}{\frac{C_{\varphi kl}\left(T\right)}{C_{\varphi kl}\left(T\left(0\right)\right)}C_{\varphi kl}^{\xi }}}=\frac{C_{\varphi kl}\left(T\right)}{C_{\varphi kl}\left(T\left(0\right)\right)}\left[C_{\varphi kl}^{\infty }+\left(C_{\varphi kl}^{0}-C_{\varphi kl}^{\infty }\right)\sum_{m=1}^{N}W_{m}exp\left(-\frac{\xi^{t}}{\tau_{m}}\right)\right]$$

Then, the thermoelastic stiffness coefficient is $d_{\varphi kl}^{T\left(t\right)}=C_{\varphi kl}\left(T\right)/C_{\varphi kl}\left(T\left(0\right)\right)$.

According to Equation (32), the stress value at time t is:(45)$$\sigma_{\varphi }^{t}=\int_{0}^{t}C_{\varphi kl}^{\xi -\xi^{\prime }}\frac{\partial \varepsilon_{kl}}{\partial \xi^{\prime }}d\xi^{\prime }$$

According to Equations (44) and (45), the stress $\sigma_{\varphi }^{t}$ at the reduced time $\xi^{\prime }$ can be rewritten as:(46)$$\sigma_{\varphi }^{t}=\int_{0}^{\xi^{\prime }}d_{\varphi kl}^{T\left(t\right)}\left[C_{\varphi kl}^{\infty }+\left(C_{\varphi kl}^{0}-C_{\varphi kl}^{\infty }\right)\sum_{m=1}^{N}W_{m}exp\left(-\frac{\xi^{t}-\xi^{\prime }}{\tau_{m}}\right)\right]\frac{\partial \varepsilon_{kl}}{\partial \xi^{\prime }}d\xi^{\prime }$$

The relaxation modulus $C_{\varphi kl}^{t}$ is expressed as:(47)$$C_{\varphi kl}^{t}=d_{\varphi kl}^{T\left(t\right)}C_{\varphi kl}^{\xi^{t}}=C_{\varphi kl}^{t}\left[C_{\varphi kl}^{\infty }+\left(C_{\varphi kl}^{u}-C_{\varphi kl}^{\infty }\right)\sum_{m=1}^{N}W_{m}exp\left(-\frac{\xi^{t}}{\tau_{m}}\right)\right]$$

Referring to (46), the stress $\sigma_{\varphi }^{t}$ at the reduced time $\xi^{t}$ is expressed as:
(48)$$\begin{array}{l} \sigma_{\varphi }^{t}=\int_{0}^{\xi^{t}}d_{\varphi kl}^{T\left(t\right)}\left[C_{\varphi kl}^{\infty }+\left(C_{\varphi kl}^{0}-C_{\varphi kl}^{\infty }\right)\sum_{m=1}^{N}W_{m}exp\left(-\frac{\xi^{t}-\xi^{\prime }}{\tau_{m}}\right)\right]\frac{\partial \varepsilon_{kl}}{\partial \xi^{\prime }}d\xi^{\prime }\\ \;\;=\int_{0}^{\xi^{t-\Delta t}}d_{\varphi kl}^{T\left(t\right)}\left[C_{\varphi kl}^{\infty }+\left(C_{\varphi kl}^{0}-C_{\varphi kl}^{\infty }\right)\sum_{m=1}^{N}W_{m}exp\left(-\frac{\xi^{t}-\xi^{\prime }}{\tau_{m}}\right)\right]\frac{\partial \varepsilon_{kl}}{\partial \xi^{\prime }}d\xi^{\prime }\\ \;\;+\int_{\xi^{t-\Delta t}}^{\xi^{t}}d_{\varphi kl}^{T\left(t\right)}\left[C_{\varphi kl}^{\infty }+\left(C_{\varphi kl}^{0}-C_{\varphi kl}^{\infty }\right)\sum_{m=1}^{N}W_{m}exp\left(-\frac{\xi^{t}-\xi^{\prime }}{\tau_{m}}\right)\right]\frac{\partial \varepsilon_{kl}}{\partial \xi^{\prime }}d\xi^{\prime } \end{array}$$

In the formula, the reduced time $\xi^{t}$ is discretized for the sum of the reduced time $\xi^{t-\Delta t}$ at time t and the reduced time increment $\Delta \xi^{t}$ at time Δt, that is:(49)$$\xi^{t}=\xi^{t-\Delta t}+\Delta \xi^{t}$$

Thus, $\Delta \xi^{t}=\int_{t-\Delta t}^{t}\frac{d\tau }{\alpha_{T}}$

The stress increment equation $\Delta \sigma_{\varphi }^{t}$ at time t is expressed as:
(50)$$\begin{array}{l} \Delta \sigma_{\varphi }^{t}=\sigma_{\varphi }^{t}-\sigma_{\varphi }^{t-\Delta t}\\ \;\;\;=\overset{H1_{\varphi }}{\overset{︷}{\int_{0}^{\xi^{\prime }}\left(d_{\varphi kl}^{T\left(t\right)}-d_{\varphi kl}^{T\left(t-\Delta t\right)}\right)C_{\varphi kl}^{\infty }\frac{\partial \varepsilon_{kl}}{\partial \xi^{\prime }}d\xi^{\prime }}}\\ \;\;+\overset{H2_{\varphi }}{\overset{︷}{\int_{0}^{\xi^{t-\Delta t}}\left(C_{\varphi kl}^{0}-C_{\varphi kl}^{\infty }\right)\sum_{m=1}^{N}W_{m}exp\left(-\frac{\xi^{t}-\xi^{\prime }}{\tau_{m}}\right)\left[d_{\varphi kl}^{T\left(t\right)}exp\left(-\frac{\xi^{t}}{\tau_{m}}\right)-d_{\varphi kl}^{T\left(t-\Delta t\right)}\right]\frac{\partial \varepsilon_{kl}}{\partial \xi^{\prime }}d\xi^{\prime }}}\\ \;\;+\overset{H3_{\varphi }}{\overset{︷}{\int_{\xi^{t-\Delta t}}^{\xi^{t}}d_{\varphi kl}^{T\left(t\right)}\left(C_{\varphi kl}^{\infty }\left(C_{\varphi kl}^{0}-C_{\varphi kl}^{\infty }\right)\sum_{m=1}^{N}W_{m}exp\left(-\frac{\xi^{t}-\xi^{\prime }}{\tau_{m}}\right)\right)\frac{\partial \varepsilon_{kl}}{\partial \xi^{\prime }}d\xi^{\prime }}} \end{array}$$

Among them, $H1_{\varphi }$ can be written in recursive form,
(51)$$H1_{\varphi }=\sum_{k}^{2}\left(d_{\varphi kl}^{T\left(t\right)}-d_{\varphi kl}^{T\left(t-\Delta t\right)}\right)A_{\varphi kl}^{t}$$
(52)$$A_{\varphi kl}^{t}=\int_{0}^{\xi^{t}}C_{\varphi kl}^{\infty }\frac{\partial \varepsilon_{kl}}{\partial \xi^{\prime }}d\xi^{\prime }=\int_{0}^{\xi^{t-\Delta t}}C_{\varphi kl}^{\infty }\frac{\partial \varepsilon_{kl}}{\partial \xi^{\prime }}d\xi^{\prime }+\int_{\xi^{t-\Delta t}}^{\xi^{t}}C_{\varphi kl}^{\infty }\frac{\partial \varepsilon_{kl}}{\partial \xi^{\prime }}d\xi^{\prime }$$

At very small time increments, the temperature T and the curing degree are extremely small and negligible, and the conversion factor is constant, so it can be expressed as:(53)$$\Delta \xi^{t}=\int_{t-\Delta t}^{t}\frac{d\tau }{\alpha_{T}}=\frac{\Delta t}{\alpha_{T}}$$

In addition,
(54)$$\frac{\partial \varepsilon_{kl}}{\partial \xi^{\prime }}\approx \frac{\Delta \varepsilon_{kl}}{\Delta \xi^{\prime }}\;\left(\xi^{t-\Delta t}\leq \xi^{\prime }\leq \xi^{t}\right)$$

Then,
(55)$$A_{\varphi kl}^{t}=\int_{0}^{\xi^{t-\Delta t}}C_{\varphi kl}^{\infty }\frac{\partial \varepsilon_{kl}}{\partial \xi^{\prime }}d\xi^{\prime }+\int_{\xi^{t-\Delta t}}^{\xi^{t}}C_{\varphi kl}^{\infty }\frac{\partial \varepsilon_{kl}}{\partial \xi^{\prime }}d\xi^{\prime }=A_{\varphi kl}^{t-\Delta t}+C_{\varphi kl}^{\infty }\Delta \varepsilon_{kl}^{t}$$

Use the same method to deal with $H2_{\varphi }$ and $H3_{\varphi }$.

Finally, the stress increment $\Delta \sigma_{\varphi }^{t}$ at time t is expressed as:(56)$$\begin{array}{l} \Delta \sigma_{\varphi }^{t}={\bar{C}}_{\varphi kl}^{*}\Delta \varepsilon_{ij}^{t}+\sum_{k}^{2}\sum_{l}^{2}\left(d_{\varphi kl}^{T\left(t\right)}-d_{\varphi kl}^{T\left(t-\Delta t\right)}\right)A_{\varphi kl}^{t}\\ \;\;\;\;\;+\sum_{k=1}^{2}\sum_{l=1}^{2}\sum_{m=1}^{N}\left[d_{\varphi kl}^{T\left(t\right)}exp\left(-\frac{\Delta \xi^{t}}{\tau_{m}}\right)-d_{\varphi kl}^{T\left(t\right)}\right]S_{\varphi kl_{m}}^{t} \end{array}$$

In addition,
(57)$${\bar{C}}_{\varphi kl}^{*}=d_{\varphi kl}^{T\left(t\right)}\left\{C_{\varphi kl}^{\infty }+\sum_{m=1}^{N}{\bar{C}}_{\varphi kl}W_{m}\frac{\alpha_{T}\tau_{m}}{\Delta t}\left[1-exp\left(-\frac{\Delta \xi^{t}}{\tau_{m}}\right)\right]\right\}$$
(58)$$A_{\varphi kl}^{t}=A_{\varphi kl}^{t-\Delta t}+C_{\varphi kl}^{\infty }\Delta \varepsilon_{kl}^{t}$$
(59)$$S_{\varphi {kl}_{m}}^{t}=exp\left(-\frac{\Delta \xi^{t}}{\tau_{m}}\right)S_{\varphi {kl}_{m}}^{t-\Delta t}+{\bar{C}}_{\varphi kl}^{*}W_{m}\frac{\alpha_{T}\tau_{m}\Delta \xi_{kl}^{t}}{\Delta t}\left[1-exp\left(-\frac{\Delta \xi^{t}}{\tau_{m}}\right)\right]$$

## 5. Verification of the Constitutive Models

To verify the constitutive models of both TSM and TCM CFRP materials, this paper selects the molded flat structure of a CFRP material based on Kim’s research [12,13]. Four layers were molded in the layering angle of [0/90/90/0]. Compared to the viscoelastic constitutive model proposed by Zocher [11], the curing degree change of the center point (0, 0, 0) of the laminate during the curing process is shown in Figure 3 below:

It can be seen from Figure 3 that the three curves are basically coincident, verifying the constitutive models given in this paper.

## 6. Conclusions

This paper studies the constitutive models of the CFRP during the compression molding process. The main content and conclusions are as follows:On the basis of the constitutive model of CFRP molding and viscoelastic theory, viscoelastic mechanics, and thermodynamics, the CFRP viscoelastic constitutive equation and incremental equation of the TSMs were established, which lays a foundation for explaining the solidification and relaxation behavior of the molding process.Considering the effects of temperature on the initial stiffness and equilibrium stiffness, the thermoelastic stiffness coefficient was introduced, and the Cartesian coordinate system was converted to the cylindrical coordinate system. Thus, the viscoelastic constitutive equation and incremental equation of the TCMs were established, and the accuracy and application range of the constitutive model were improved while the calculation scale was reduced.On the basis of the viscoelastic constitutive model proposed by Zocher, the correctness and validity of the constitutive models for TSMs and TCMs were verified by comparing the change in curing degree under the anisotropic viscoelastic constitutive model.

## Figures and Tables

**Figure 1 materials-13-02277-f001:**
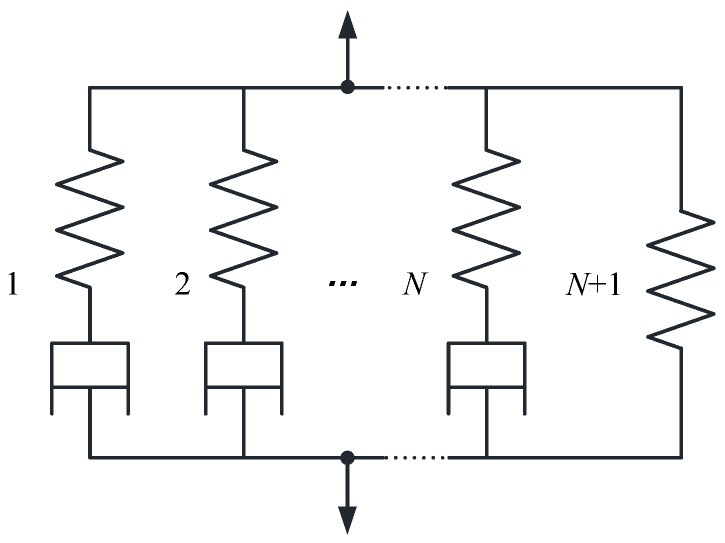
Generalized Maxwell model.

**Figure 2 materials-13-02277-f002:**
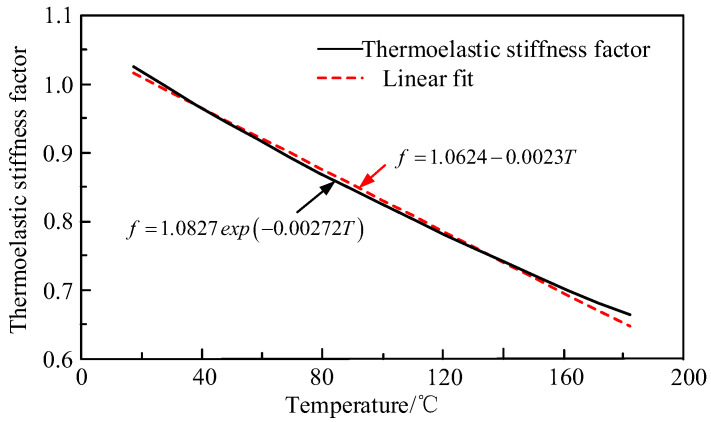
The thermoelastic stiffness coefficient of an epoxy resin.

**Figure 3 materials-13-02277-f003:**
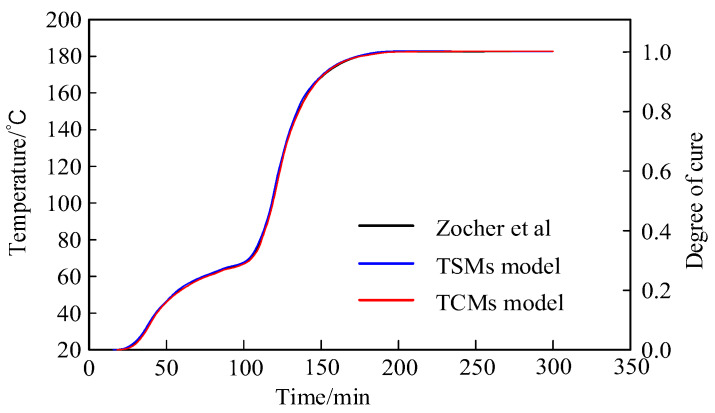
Comparison of the changes of (0,0,0) curing degree.

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
