# Peer review of "Research on Anisotropic Viscoelastic Constitutive Model of Compression Molding for CFRP"

_materials, 2020, doi:10.3390/ma13102277_

Round 1
Reviewer 1 Report
The article covers the topic of the Anisotropic Viscoelastic Constitutive Model of Compression Molding for CFRP. The topic and the supporting research are informative and present added value to the body of knowledge on the CFRP area. The topic of the article is in scope of journal. The authors thoroughly analyzed CFRP mechanics. All equations relevant for
the preparation of the models were shown in this paper.
In addition, constitutive models have been positively verified. Some following modification should be considered:
1. I suggest to add point 2 - Research significance - Please descibe here the main essence of the experiment (especially text in lines 69-78).
I suggest to pay attention on the information for readers - why your researches are so crucial for future developing of CFRP?
2. It would be beneficial if the authors provided basic information about the CFRP production process.
Author Response
Point 1: Research significance - Please descibe here the main essence of the experiment (especially text in lines 69-78). I suggest to pay attention on the information for readers - why your researches are so crucial for future developing of CFRP?
Response 1: In this part, the author has further elaborated the significance of this article.
Point 2: It would be beneficial if the authors provided basic information about the CFRP production process.
Response 2: The carbon-fiber-reinforced Polymer molding process production is the basic content, the difficulty is to choose different processing technology according to different materials, such as: pressure, holding temperature, holding time, opening temperature, etc. , therefore, the research content of this paper will lay a theoretical foundation for the formulation of the molding process.
Reviewer 2 Report
The article, in general, is correct although its contributions are not very relevant.
- Both in the abstract and at the beginning of the introduction it is written that carbon fibre is the king of 21st century materials. The statement is not fortunate. It should, in any case, be said that carbon fibre is one of the most relevant materials from the end of the 20th century to the present day.
- In line 46, it is written "Scholars at home and abroad" should, in any case, say "Scholars worldwide"
- The text from line 124 to line 145 is too basic to appear in an paper in this journal.
- The conclusions are not sufficiently worked out and must be rewritten.
Author Response
Point 1: Both in the abstract and at the beginning of the introduction it is written that carbon fibre is the king of 21st century materials. The statement is not fortunate. It should, in any case, be said that carbon fibre is one of the most relevant materials from the end of the 20th century to the present day.
Response 1: The way of expression proposed by the peer reviewers is more accurate and has been revised in accordance with the comments.
Point 2: In line 46, it is written "Scholars at home and abroad" should, in any case, say "Scholars worldwide"
Response 2: The way of expression proposed by the peer reviewers is more accurate and has been revised in accordance with the comments.
Point 3: The text from line 124 to line 145 is too basic to appear in an paper in this journal.
Response 3: The section has been modified for the context and the text of too basic proposed by the reviewers has been removed.
Point 4: The conclusions are not sufficiently worked out and must be rewritten.
Response 4: The author has made further supplement and improvement to the conclusion content, and has be rewriting.

Reviewer 3 Report
The paper contains a brief introduction about developed constitutive models, theoretical basis of anisotropic linear elastic mechanics, anisotropic viscoelastic mechanics as well as thermodynamics. Then two new constitutive models for the thermorheologically simple CFRP materials and thermorheologically complex materials are described. Generally the structure of the paper is clear and understandable, but the difference between TSMs and TCMs should be explained in the introduction. The sections 3. and 4. contains only descriptions of the models, without deeper discussion why particular components of the models were applied and how they influence on the results. This part of the paper can be possibly improved. In the section 5. verification of the models is presented. The comparison was done only with one other constitutive model (Zocher) and one parameter. Did the Authors make any experimental tests to verify their models? Comparison with experimental data would be beneficial. Moreover the curve of Zocher model in Figure 3 should be drawn with different colour (not black), because it is not visible at the beginning of the plot. Citation to Zocher model in Figure 3 should be added.
Typos:
Line 12: polymer should start from small letter
line 100, 104, 106: double spaces,
lines 97-98: various commas
line 188: should be Figure 1
lines 259, 328, 329: please provide the citations (Kim, Zocher)
lines 278-279: “this paper uses…”?
line 335: dot at the end of the sentence
Line 345: should be: “Thus, the…”
Punctuation should be improved: i.e.: spaces/commas: lines 4, 362, introduction: citations (lack of spaces before square brackets), unnecessary spaces inside round brackets.
Author Response
Point 1: Generally the structure of the paper is clear and understandable, but the difference between TSMs and TCMs should be explained in the introduction.
Response 1: The difference between TSMs and TCMs are are described in part 4. The CFRP constituent model of the TCMS-4.1 thermal elastic expressions of the TCMS.
Point 2:The sections 3. and 4. contains only descriptions of the models, without deeper discussion why particular components of the models were applied and how they influence on the results. This part of the paper can be possibly improved.
Response 2: The authors are currently conducting theoretical analysis and experiments to further explain the mechanism of the particular components of the models were applied and influence on the results ,and the results will be published in a subsequent paper.
Point 3: In the section 5. verification of the models is presented. The comparison was done only with one other constitutive model (Zocher) and one parameter. Did the Authors make any experimental tests to verify their models? Comparison with experimental data would be beneficial.
Response 3: Many researchers have verified the model established by Zocher, and the results show that the model established by Zocher is basically consistent with the experimental results, so in this paper, The authors are does not do further experimental verification, but as a standard.
Point 4: Moreover the curve of Zocher model in Figure 3 should be drawn with different colour (not black), because it is not visible at the beginning of the plot.
Response 4: He curve of Zocher model in Figure 3 is the one that exists in other papers. So, solid black lines are chosen, but because the three curves are so close, solid black lines are not obvious.
Point 5:Typos
Response 5: Thanks to the reviewers for their questions about the details, which have been revised.
